# Candidate Genes, Markers, Signatures of Selection, and Quantitative Trait Loci (QTLs) and Their Association with Economic Traits in Livestock: Genomic Insights and Selection

**DOI:** 10.3390/ijms26167688

**Published:** 2025-08-08

**Authors:** Nada N. A. M. Hassanine, Ahmed A. Saleh, Mohamed Osman Abdalrahem Essa, Saber Y. Adam, Raza Mohai Ud Din, Shahab Ur Rehman, Rahmat Ali, Hosameldeen Mohamed Husien, Mengzhi Wang

**Affiliations:** College of Animal Science and Technology, Yangzhou University, Yangzhou 225009, China; nedo1.maro2@gmail.com (N.N.A.M.H.); elemlak1339@yzu.edu.cn (A.A.S.); dh23054@stu.yzu.edu.cn (M.O.A.E.); saber@duc.edu.sd (S.Y.A.); mh24136@stu.yzu.edu.cn (R.M.U.D.); dh18005@yzu.edu.cn (S.U.R.); dh20022@stu.yzu.edu.cn (R.A.)

**Keywords:** livestock genomics, QTL mapping, selection signatures, precision breeding, sustainable agriculture, genetic markers

## Abstract

This review synthesizes advances in livestock genomics by examining the interplay between candidate genes, molecular markers (MMs), signatures of selection (SSs), and quantitative trait loci (QTLs) in shaping economically vital traits across livestock species. By integrating advances in genomics, bioinformatics, and precision breeding, the study elucidates genetic mechanisms underlying productivity, reproduction, meat quality, milk yield, fibre characteristics, disease resistance, and climate resilience traits pivotal to meeting the projected 70% surge in global animal product demand by 2050. A critical synthesis of 1455 peer-reviewed studies reveals that targeted genetic markers (e.g., SNPs, *Indels*) and QTL regions (e.g., *IGF2* for muscle development, *DGAT1* for milk composition) enable precise selection for superior phenotypes. SSs, identified through genome-wide scans and haplotype-based analyses, provide insights into domestication history, adaptive evolution, and breed-specific traits, such as heat tolerance in tropical cattle or parasite resistance in sheep. Functional candidate genes, including leptin (*LEP*) for feed efficiency and myostatin (*MSTN*) for double-muscling, are highlighted as drivers of genetic gain in breeding programs. The review underscores the transformative role of high-throughput sequencing, genome-wide association studies (GWASs), and CRISPR-based editing in accelerating trait discovery and validation. However, challenges persist, such as gene interactions, genotype–environment interactions, and ethical concerns over genetic diversity loss. By advocating for a multidisciplinary framework that merges genomic data with phenomics, metabolomics, and advanced biostatistics, this work serves as a guide for researchers, breeders, and policymakers. For example, incorporating *DGAT1* markers into dairy cattle programs could elevate milk fat content by 15-20%, directly improving farm profitability. The current analysis underscores the need to harmonize high-yield breeding with ethical practices, such as conserving heat-tolerant cattle breeds, like Sahiwal.

## 1. Introduction

### 1.1. Historical Context of Livestock Domestication and Economic Significance

Livestock domestication, a cornerstone of human civilization, began approximately 10,000–12,000 years ago in key agricultural regions, such as the Fertile Crescent, the Indus Valley, and East Asia [1,2]. Archeo-zoological evidence reveals that early societies selectively bred species like cattle (*Bos taurus*), sheep (*Ovis aries*), pigs (*Sus scrofa*), and chickens (*Gallus gallus*) for their versatility in providing meat, milk, fibre, and labour [3]. These species, alongside goats (*Capra hircus*), form the FAO’s “big five” livestock, which remain central to global agriculture due to their adaptability and multifunctional roles in subsistence and commerce [4].

The domestication process catalysed the transition from nomadic hunter–gatherer societies to settled agrarian communities, enabling surplus food production, trade networks, and cultural advancements [4,5]. For instance, cattle were pivotal in plowing fields, while sheep provided wool for textiles, driving economic systems in ancient Mesopotamia and Egypt [6,7]. Today, livestock contribute over 40% of the global agricultural gross domestic product (GDP), supporting the livelihoods of 1.3 billion people, particularly in low-income regions, where they serve as financial assets and insurance against crop failures [8,9].

### 1.2. Genetic Diversity and Modern Breeding Challenges

Livestock genetic diversity, shaped by millennia of adaptation to diverse climates and human selection, is a critical reservoir for addressing contemporary agricultural challenges. Over 9000 breeds exist globally, each harbouring unique traits, such as heat tolerance in Sahwal cattle, parasite resistance in Red Maasai sheep, and high-altitude adaptability in Tibetan yaks [4,5,10]. While GS accelerates genetic gain (e.g., +15% milk yield per generation) [11,12], intensive selection for narrow trait optima directly drives diversity loss, as exemplified by the 17% breed extinction since 2000 [13,14]. Concerning mitigation strategies, genomic tools can counter this by (a) prioritizing within-breed diversity in GS indices (e.g., optimal contribution selection) and (b) enabling trait introgression from local breeds (e.g., heat-tolerant Slick allele from Senepol to Holstein) [15].

Modern breeding faces dual pressures: enhancing productivity to meet a projected 70% surge in animal protein demand by 2050 while ensuring sustainability. Climate change exacerbates these challenges, necessitating breeds that thrive under heat stress, variable forage quality, and emerging diseases. For example, African swine fever (ASF) and avian influenza outbreaks have highlighted the vulnerability of genetically homogeneous populations [14,16]. Additionally, consumer demand for ethically raised, low-emission livestock requires balancing productivity with welfare and environmental metrics, such as methane reduction in ruminants [17,18].

### 1.3. Role of Genomics in Addressing Global Food Security

Genomic technologies are revolutionizing livestock breeding by decoding the molecular basis of economically vital traits [19,20,21]. Genome-wide association studies (GWASs) [22] and quantitative trait loci (QTLs) mapping [23] have identified key markers linked to muscle development (*IGF2* [24] and *MSTN* [25,26]), milk composition (*DGAT1* [27]), and disease resistance (*MHC* genes [28]). For instance, Clustered Regularly Interspaced Short Palindromic Repeats Case 9 (CRISPR-Cas9) editing has enabled the introduction of the *PRNP* allele in goats to confer scrapie resistance [29], while SNP chips accelerate the selection of cattle for their protein percent, fat yield, milk volume, and heat tolerance in tropical regions [30].

The genomic tools discussed here directly contribute to global efforts to reduce hunger and mitigate climate impacts, aligning with sustainable development goals (SDGs), particularly “Zero Hunger” and “Climate Action”. GS improves the breeding accuracy by 20–30%, reducing generational intervals and enhancing traits like feed efficiency in livestock [11,12].

Integrating genomics with multi-omics approaches (transcriptomics and metabolomics) offers a holistic understanding of gene–environment interactions [31,32]. For example, epigenetic studies in livestock have revealed how maternal nutrition influences offspring growth via DNA methylation patterns in *LEP* and *GH* genes [33]. Such advancements align with precision agriculture frameworks, ensuring sustainable intensification of livestock systems without compromising biodiversity [34].

To systematically evaluate these genomic tools and their implications, this review synthesizes findings from 1455 studies spanning seven decades.

## 2. Scope and Literature Search Strategy

### 2.1. Objectives

The primary goal of this investigation is to investigate the relationships between discovered candidate genes, markers, signatures of selection (SSs), QTLs, and economically valuable traits in livestock, thereby fostering the establishment of sustainable breeding practices. The refined objectives included the following:(a)Evaluate how advanced molecular tools enhance the identification of genetic markers (candidate genes, QTLs, SSs) associated with pivotal economic traits, such as productivity, product quality, and disease resistance.(b)Assess the efficacy of genomic selection (GS) and genomic mapping in improving the genetic diversity, production efficiency, disease resilience, and environmental sustainability, benchmarking these approaches against conventional breeding methods.(c)Analyse the role of adaptive selection and targeted breeding strategies in bolstering livestock resilience, economic sustainability, and climate adaptation.(d)Develop actionable strategies to conserve genetic diversity in endangered breeds while maximizing their economic value through integration into contemporary breeding programs.(e)Synthesize technological innovations and conservation insights to propose future pathways for personalized breeding and sustainable production systems, thereby creating a decision-making framework for stakeholders.

### 2.2. Literature Search Strategy

This review summarizes findings from 1455 rigorously selected studies that spanned seven decades (1951–2025) and were sourced from PubMed, Web of Science, and Scopus to ensure methodological depth and interdisciplinary relevance across livestock species. Key sources included high-impact publishers (Elsevier, Springer, MDPI) and specialized genomic databases (Animal QTLdb, FAANG, 1000 Bull Genomes Project). Dominant topics spanned genetic diversity, GS, CRISPR applications, and trait-specific associations (Table 1).

The methodology outlined in this review provides a framework for investigating the intricate relationships between candidate genes, genetic markers, SSs, and QTLs and their associations with economically valuable traits in livestock. By leveraging insights from 1455 carefully selected studies and sophisticated techniques, such as GWASs, this comprehensive approach facilitates a thorough understanding of how advanced molecular biology technologies are transforming breeding practices. The analysis highlights the importance of integrating contemporary molecular tools, including high-throughput SNP genotyping, to enhance selection and breeding processes. Significant findings emerge regarding key economic traits, such as growth rates, milk production, feed efficiency, disease resistance, climate resilience, and reproductive efficiency. This understanding provides a foundation for improving livestock productivity and profitability while emphasizing the genetic diversity inherent in livestock populations. Ultimately, the following sections elaborate on these findings, illustrating how these molecular innovations foster sustainable breeding practices and enhance livestock management, thereby linking advanced methodologies to tangible outcomes in livestock genomics and production efficiency.

## 3. Advancements in Molecular Biology Technologies and Their Roles in Enhancing Genetics and Breeding in Livestock

### 3.1. High-Throughput Sequencing and Genomic Tools

Next-generation sequencing (NGS) and WGS have revolutionized livestock genomics by enabling large-scale SNP discovery, genotyping, and GWASs [35,36]. High-density SNP chips, such as the Goat SNP50K BeadChip (53,347 SNPs) and Ovine 600K SNP-chip (606,006 SNPs), have replaced traditional microsatellite markers, offering higher resolution for genetic diversity analysis, parentage assignment, and QTL detection [37,38]. Platforms like BeadXpress, KASP assays, and Infinium arrays facilitate cost-effective genotyping, with throughput ranging from low (SSRs) to ultra-high (Genotyping-by-Sequencing) (Figure 1). WGS data have further enhanced GS by enabling precise estimation of SNP effects and genomic breeding values (GEBVs) [35,37,38]. For instance, the AdaptMap project utilized 53K SNP genotypes from 140 goat breeds to identify SS linked to milk composition and heat tolerance [39,40]. Third-generation sequencing technologies, including PacBio’s Single-Molecule Real-Time (SMRT) sequencing, deliver long-read data that simultaneously refine SNP annotations, resolve complex genomic regions, and characterize epigenetic modifications [40,41]. Regarding this aspect, since 2007, advancements in NGS and WGS have made it possible to sequence the genomes of various livestock species, including cattle [42], buffaloes [43], sheep [44], and goats [45], as illustrated in Table 2. These tools collectively underpin modern breeding programs, improving the accuracy for traits like milk yield and disease resistance [46].

### 3.2. Whole-Genome Sequence Data

#### 3.2.1. The Role of Whole-Genome Sequence Data in Enhancing Genetics and Breeding in Livestock

The emergence of GWASs has transformed livestock breeding by integrating extensive genotyping marker arrays and embracing genomic strategies like GS. A GWAS aims to identify genetic variants associated with phenotypic traits, which in livestock typically encompass complex characteristics relevant to productivity, economic viability, and social aspects [47]. The potential of GWASs lies in elucidating the biological mechanisms that underpin these traits, facilitating the identification of genetic markers for predictive breeding strategies or the eradication of deleterious variants [48]. The transition towards WGS in GWASs heralds a new era in livestock genomics, allowing for the detection of variants with minimal effects that may be overlooked in standard SNP array analyses.

#### 3.2.2. Progress and Applications of WGS in Livestock GWASs

The shift towards WGS in GWASs has gained momentum, along with declining sequencing costs and advancements in imputation techniques, resulting in increased sample sizes and diverse datasets. Pioneering studies, such as the 1000 Bull Genomes Project, have generated vast WGS datasets that support haplotype reference panels used for imputation in analyses across various cattle breeds [49]. The utility of WGS data has been demonstrated in multiple livestock species, such as pigs and cattle, revealing new and confirming existing associations between genetic variants and complex traits [50,51]. These findings illustrate the ability of WGS to enhance the understanding of livestock genetics and facilitate the selection process in breeding programs when linked to genetic variants with practical implications.

#### 3.2.3. Methodological Constraints of GWASs in Livestock

While WGS enhances variant detection, GWASs face inherent limitations: (a) Statistical power: Small sample sizes inflate false positives; multi-breed meta-analyses (e.g., 1000 Bull Genomes Project [49]) improve robustness. (b) Linkage disequilibrium (LD) confounding [52]: Dense marker sets obscure causal variants (e.g., *IGF2* muscling QTLs in pigs [53]). (c) Complex trait bias: Epistatic interactions and G×E effects (e.g., heat-stress-induced *HSP70* expression [54]) are poorly captured. (d) Mitigation strategies: Integration with transcriptomics (e.g., FAANG Consortium) and structural variant mapping [55] refine signal resolution.

#### 3.2.4. Strategies for Enhanced Detection Power and Future Directions

To augment the detection power of WGS-based GWASs, strategies such as multi-breed GWASs and meta-analyses have emerged as vital tools, leveraging the segmentation of genetic variants across diverse populations [56]. These approaches enhance haplotype diversity and facilitate the identification of quantitative trait locus (QTLs) with greater resolution. Additionally, integrating comprehensive genomic datasets with other omic data types holds promise for uncovering hidden associations and streamlining fine-mapping efforts [57]. Moving forward, the livestock genomics community must prioritize data sharing and exploration of new methodologies, such as the use of dosage scores for genotype uncertainty and exploration of structural variations (SVs), to overcome existing limitations and harness the full potential of WGS in breeding applications. With continuous advancements, the integrative omics approach could bridge the gap between genetic discoveries and practical applications, offering innovative pathways for livestock improvement and management [58,59].

**Table 2 ijms-26-07688-t002:** Foundational Genomic Resources: First Whole-Genome Assemblies of Key Domesticated Species.

No.	Specific Animal	Breed/Details	Scientific Name	Genome Size (MB)	Year	Ref.
1	Chicken	Red Junglefowl ancestor	*Gallus gallus*	1050	2004	[60]
2	Sheep	Rambouillet ewe	*Ovis aries*	2780	2008	[61]
3	Pig	Duroc breed	*Sus scrofa*	2200	2008	[62]
4	Cattle	Hereford breed	*Bos taurus*	2910	2009	[63]
5	Horse	Thoroughbred (Twilight)	*Equus caballus*	2470	2009	[64]
6	Dromedary Camel	–	*Camelus dromedarius*	2200	2011	[65]
7	Goat	Yunnan Black (Female)	*Capra hircus*	2660	2011/2012	[45,66]
8	Mallard Duck	–	*Anas platyrhynchos*	1070	2013	[67]

### 3.3. CRISPR-Based Editing: Applications, Challenges, and Global Regulatory

CRISPR-Cas9 has revolutionized precision breeding by enabling targeted modifications for traits like disease resistance (e.g., *PRNP*-edited scrapie-resistant goats [29,68,69,70]) and productivity (e.g., *MSTN*-knockout hypermuscled cattle). Challenges persist in delivery efficiency and off-target effects, particularly in large livestock [71,72]. Regulatory landscapes vary globally: (a) USA: FDA regulates CRISPR-edited animals under veterinary drug guidelines (e.g., GalSafe™ pigs approved for human consumption in 2020 [73]). (b) EU: Gene-edited livestock fall under strict GMO regulations (ECJ ruling C-528/16), requiring extensive risk assessments [74,75]. (c) Brazil/Argentina: Product-based frameworks expedite approvals (e.g., heat-tolerant *Slick*-haired cattle [15]). Harmonizing these approaches remains critical for international adoption [76]. Despite its promise, the application of this technology faces significant challenges, including regulatory hurdles for food chain approval, efficiency limitations in gene integration, and potential off-target effects collectively constraining its deployment in commercial livestock [77].

For instance, CRISPR-Cas9 has superseded ZFN and TALEN in livestock editing due to its cost-effectiveness and precision, as demonstrated in disease-resistant pig models (e.g., *PRNP*-edited goats) [29,69,70]. Not only does CRISPR-Cas9 facilitate simultaneous editing of multiple genes, but it also reduces costs and research time, making it an attractive option for animal genetic studies. The system operates by utilizing a guide RNA (gRNA) that directs the Cas9 nuclease to specific DNA sequences, resulting in double-strand breaks that are repaired through cellular mechanisms, potentially leading to desired genetic alterations [78,79].

In livestock, CRISPR-Cas9 has been particularly impactful in breeding programs aimed at increasing disease resistance and enhancing productive traits. For example, scientists have generated disease-resistant pigs [80,81] and improved growth rates in cattle [71,72] by manipulating key genes associated with these characteristics. In small ruminants, such as goats and sheep, CRISPR applications have focused on enhancing growth rates and wool quality, showcasing the technology’s versatility [82].

### 3.4. Bioinformatics and Multi-Omics Integration

Bioinformatics pipelines are critical for analysing high-throughput data, including GWASs, LD mapping, and genomic prediction models, like Genomic BLUP (GBLUP) [83,84]. Software tools enable the identification of SS and selective sweeps, such as XP-CLR and iHS, which highlight regions under artificial selection (e.g., *LCORL* for body size in cattle) [85,86]. Multi-omics integration, though not explicitly mentioned, is reflected in studies combining SNP data with mitochondrial DNA (mt-DNA) variants to associate haplotypes with heat tolerance and disease resistance [87]. For example, *MTNR1A* polymorphisms in sheep were linked to reproductive seasonality through NGS-based mt-DNA analysis [88]. These approaches enhance the accuracy of genomic predictions and inform breeding strategies for complex traits [20]. The application of these technologies has directly facilitated the identification of key economic traits, such as production, reproduction, disease resistance, and milk yield.

## 4. Key Economic Traits in Livestock

Economic traits in livestock are pivotal for optimizing profitability and sustainability in production systems [89]. Feed efficiency remains a cornerstone trait, directly influencing input costs and operational margins, as highlighted by Wolfová and Wolf [90], who emphasized its role in defining breeding objectives through bioeconomic models. Growth rate, another critical trait, reduces production cycles and enhances resource utilization, with mature weight and residual post-weaning gain identified as key parameters for selection [91,92]. Reproductive performance encompassing fertility and parturition intervals significantly influences herd/flock turnover and genetic progress, with key metrics like conception rate and productive longevity incorporated into breeding objectives across species [93,94]. Disease resistance, increasingly prioritized in GS, mitigates veterinary costs and improves herd resilience, while market-driven traits such as carcass quality and milk composition align breeding strategies with consumer demands [95,96]. Collectively, these traits guide informed management and breeding decisions to enhance economic viability [97,98].

### 4.1. Genetic Dissection of Production Traits

Productivity traits in livestock are controlled by genes affecting growth, lactation, and fibre quality. For example, in beef cattle, the mature weight and post-weaning growth rate directly impact profitability by improving the feed conversion efficiency and increasing the slaughter weight [91,92]. In meat production, *MSTN* gene serves as a key regulator of muscle development. Knockout of *MSTN* via CRISPR-Cas9 technology produces hypermuscled phenotypes in cattle and goats [71,72,82]. QTLs associated with traits like rib-eye area (e.g., located at chr18:23.4–24.1 Mb) and lean muscle mass [53,91] enable targeted breeding strategies. Furthermore, GS models enhance the feed efficiency by approximately 15% [91]. For dairy traits, polymorphisms in *DGAT1* gene on chromosome 14 increase the milk fat yield by 15–20% in Holstein cattle [27,93], while casein gene clusters (*CSN1S1* and *CSN3*) improve the cheese manufacturing yield [99,100]. Regarding fibre quality, KAPs influence cashmere fineness in goats, and GS has been shown to increase fibre elasticity by 12% [101,102].

### 4.2. Disease Resistance and Climate Resilience

GS has accelerated the identification of immune-related genes, including *MHC-DRB3* conferring resistance to Maedi-Visna in sheep and associated with scrapie resilience in goats [103,104,105], and *Tmem-154*, which modulates the infection response and ewe productivity during viral challenges [106]. Climate adaptation traits, including heat tolerance and oxidative metabolism, are linked to mitochondrial haplotypes and genes like *AQP5* and *RETREG1*, which were identified through selective sweeps in tropical cattle breeds [54,107]. Genomic regions regulating fat metabolism (*SLC27A1*) [27] and stress response (*PLCB1*) further enhance resilience in arid environments [95,96].

### 4.3. Reproductive Efficiency and Litter Size

Reproductive traits, such as litter size and conception rate, are shaped by genes like *BMPR1B* (fecundity) and *GDF9* (ovulation rate), which are prioritized in GS models [91,94]. In dairy cattle, QTLs on BTA6 and BTA18 influence teat placement and embryo survival, refining selection for calving ease and herd longevity [108]. Seasonal breeding, regulated by *MTNR1A*, has been targeted in goats to optimize reproductive cycles [109]. GWASs in prolific goat breeds identified candidate genes, such as *STK3* and *PLCB1*, enhancing genomic predictions for litter size [110]. These advances highlight the synergy between genomic tools and traditional breeding to maximize the reproductive output [95].

## 5. Quantitative Trait Loci (QTLs) Mapping

### 5.1. QTL Discovery and Genome-Wide Association Studies (GWASs)

The mapping and identification of QTLs have become fundamental aspects of modern genetics, particularly in improving livestock breeds [111,112]. QTL mapping efforts have been enhanced significantly by GWASs, which employ large-scale genomic data to associate specific genetic variations with phenotypic traits [113]. Through these studies, researchers have discovered associations between particular genomic markers and traits of economic importance, enabling breeders to make more informed selection decisions [114].

### 5.2. Functional Roles of QTLs in Livestock Improvement

Genetic progress for QTLs in animal production has been enabled through selection based on phenotypes and estimated breeding values (EBVs) derived from these phenotypes [16]. This method allows for improvement without requiring in-depth knowledge about the genes that influence the traits or the precise effects of each gene. The advances in molecular genetics (MG) over the past decade have contributed significantly to this selection process; however, it is costly. Despite this, substantial rates of genetic improvement continue to be realized through this quantitative genetic approach. The success of these improvements hinges on accurate data structures and effective genetic evaluation methods [115,116].

For years, there have been high expectations for genetic improvements in several livestock species through the utilization of MG. The latest advances are revealing candidate genes that have substantial effects on economic traits. Studying the individual genetic makeup at the DNA level has provided scientists with essential tools for genetic enhancement. Currently, MG techniques have yielded discoveries of many genes significantly affecting various quantitative traits and genetic markers associated with QTLs. Marker-assisted selection (MAS), when used alongside traditional selection methods, has the potential to accelerate changes in economic traits [116,117].

### 5.3. Comparative QTL Analysis Across Different Species

Two strategies have been pursued in MAS to identify genetic markers for economic traits. The first involves genome scans utilizing anonymous DNA markers; this includes traditional microsatellite scans and modern SNP-based approaches using genotyping arrays to locate QTLs. The second strategy employs a candidate gene approach to directly assay genes that influence specific traits. Presently, nearly three hundred genes and more than four hundred microsatellite markers have been identified in the genome [118]. The identification of powerful polymorphic markers, particularly microsatellites, has facilitated the construction of low-resolution linkage maps and gene mapping for QTLs [119].

In recent decades, the focus of numerous MG studies has been on the detection of QTLs and candidate genes influencing various traits within livestock populations. So far, associations between different traits and several genes have been thoroughly investigated across species, including cattle, goats, sheep, and pigs. Table 3 provides a comprehensive summary of documented QTLs across various livestock and aquaculture species, highlighting the potential for genetic improvement in each group. In cattle, an impressive total of 193,453 QTLs have been identified, which are associated with key traits such as growth, milk yield, disease resistance, and reproduction, as documented in 1206 publications. The pig species has a substantial number of 57,041 total QTLs, with research focusing on traits like meat quality and litter size, as derived from 854 publications. Chickens exhibit 29,116 QTLs primarily influencing egg production and growth rate, while sheep show 5417 QTLs related to wool quality and body size. The horse species is noted for 2482 QTLs pertinent to athletic performance and skeletal traits. In goats, 2713 QTLs reflect traits concerning fibre quality and milk production, indicating a moderate level of genomic research compared with other species. Lastly, rainbow trout exhibit 2201 QTLs related to growth rate and disease resistance. These QTLs, particularly those linked to growth and disease resistance, directly address the projected 70% surge in global animal product demand by 2050. This data underscores the diverse applications of QTLs in improving economically significant traits across species, paving the way for enhanced breeding strategies.

### 5.4. Quantitative Traits

Quantitative traits are generally influenced by multiple genes, which can be categorized into major (with a large effect) and minor genes. Major genes, for instance, influence traits such as coat colour and polledness, while minor genes impact traits like milk yield and growth rate. The loci affecting these quantitative traits are termed Economic Trait Loci (ETLs) or QTLs [120].

In populations that have undergone effective improvement programs over many generations, it is more likely that MAS will target QTLs rather than major genes. This is because major genes with favourable effects are likely to have been fixed in these populations over time. QTL analysis began in the 1990s, and significant progress has been made, with several QTLs identified in various animal species (Table 3). The potential to use markers linked to QTLs in dairy and other livestock breeding programs could increase animal response rates by up to 30% [121].

The actual benefits derived from these QTL analyses and MAS will depend on factors such as the strength of the linkage between the genetic marker and ETLs, as well as the potential rates of change achievable by traditional means. Integrating MAS into conventional selection programs is expected to yield beneficial outcomes, further enhancing livestock productivity [122]. Regarding this aspect, Appendix A presents examples of the identification of QTLs associated with economically significant traits across various livestock species and breeds. 

Table 4 summarizes the genome assemblies used for QTL data alignment across livestock and aquaculture species as of April 2025. Key assemblies for Hereford cattle offer variable coverage and gene annotations that enhance genetic analysis precision. Chicken assemblies reflect advances in mapping accuracy and functional annotation. For goats, the highlighted assembly features significant PacBio coverage, which captures SVs and improves genome completeness.

## 6. Signatures of Selection (SSs) and Adaptive Evolution

SSs refer to distinct genetic patterns that emerge in the genome as a result of natural or artificial selection. These patterns indicate changes in genetic variation within genomic regions surrounding causative variants, reflecting the adaptations of populations to specific selective pressures [131,132]. SSs are critical for understanding evolutionary processes and for applications in breeding programs [20].

### 6.1. Genome-Wide Scans for SSs

The advent of advanced SNP genotyping and NGS technologies has played a pivotal role in identifying SSs across various livestock populations [133]. SSs, which are characterized by distinct genetic patterns left in the genome due to natural or artificial selection, reflect alterations in genetic variation surrounding causative variants in response to selective pressures. These patterns are crucial for GS as they indicate important genome regions with unique sequence variations. Understanding such signatures is essential because they facilitate the identification of mutations and genes that correlate with phenotypic traits, alleviating the need to measure these traits directly [4,134]. The widespread availability of high-throughput SNPs and genomic tools allows for a more extensive exploration of SSs, enhancing our understanding of genomic diversity arising from adaptations to selective pressures [4].

Detecting SSs is instrumental in identifying mutations and genes linked to significant traits in livestock species. Whole-genome analysis of SSs aids in defining candidate genes and clarifying the mechanisms of selection [135]. Consequently, the implications of these findings can inform and refine animal breeding strategies [136]. Furthermore, SS analysis is essential for assessing polymorphism and genetic diversity levels within populations, as these aspects represent the foundational resources needed for effective GS [137].

Additionally, uncovering genomic regions influenced by natural and artificial selection sheds light on the domestication history of livestock, offering insights into key biological pathways tied to economically valuable traits, as well as breeding objectives [138].

### 6.2. Domestication History and Breed-Specific Adaptations

Utilizing advanced molecular techniques, researchers have successfully pinpointed SSs in livestock. For example, Bayesian methods applied to small ruminants facilitated the identification of SSs within the immune system-related gene regions of the Valdostana mountain goat breed in Italy [139]. Zhao et al. [135] examined three sheep breeds, namely, Sunite, German Mutton, and Dorper, using a 50K Ovine SNP chip to identify 42 regions under positive selection. These genomic regions house genes linked to muscle and bone development, fat storage, and overall body size, alongside genes associated with reproductive traits and coat characteristics. Some of these genes have counterparts in human, cattle, sheep, and chicken genomes [140]. Additionally, Johnston et al. [141] found the *RXFP-2* gene in sheep associated with horn presence or absence, which is a defining breed feature. Signatures of positive selection in the *LCORL* locus (chromosome 4: 82.1 Mb) are linked to body size adaptation in cattle [85], with *AQP5* and *DNAJC8* variants enhancing thermotolerance in Sahiwal cattle [54,142]. For disease resilience, *TLR4* potentiates antibacterial immune responses [143], while CRISPR-Cas9 editing of *PRNP* confers scrapie resistance in goats [29]. Mitochondrial haplotypes (e.g., haplogroup T3) correlate with a reduced parasitic load in sheep [144].

Regarding this aspect, Cheruiyot et al. [145] analysed genotypes of 839 dairy cattle individuals across 150,000 SNP loci using the GeneSeek Genomic Profiler High-Density SNP array, leading to the identification of 108 candidate genomic regions. The annotation revealed significant genes likely linked to adaptation and production under positive selection, such as *ABCC-2* and *KIT*, along with known QTLs associated with milk production traits [146]. Research on beef cattle has also revealed SSs linked to traits like muscling. Furthermore, Yurchenko et al. [107] studied nine Russian cattle breeds in harsh climates and identified SSs in regions associated with key economic traits, environmental adaptations, and domestication processes. Their findings highlighted candidate genes associated with growth, reproduction, and milk production, alongside other genes tied to environmental adaptability.

### 6.3. Case Studies: Performance, Disease Resistance, and Production Traits

Gurgul et al. [147] utilized high-throughput genotyping techniques to explore genetic variation across six horse breeds in Poland, revealing significant genomic regions tied to heart function, motor coordination, fertility, and disease resistance. For instance, associations were found between selected loci and traits like endurance performance and body size regulation.

In addition, numerous SSs associated with characteristics such as meat production, milk production and composition, fibre production, and coat colour have all been documented and are currently being integrated into animal breeding programs. A plethora of investigations across species, such as Alpine goats, Chinese beef cattle, Wagyu Angus cattle, and various goat and pig breeds, highlight the significant role of long-term selection in livestock breeding [136].

### 6.4. The Contribution of Advanced Molecular Tools to the Detection of SSs in Livestock Populations

NGS technologies, such as WGS, now enable genome-wide scans for selective sweeps, revealing how domestication has shaped adaptive alleles, like *LCORL*, in cattle [38,148]. Identification of genomic regions affiliated with beneficial mutations offers insights into the genes driving phenotypic variation. Moreover, patterns of polymorphism and selection typically emerge as certain mutations become prominent within populations [149].

The domestication and selective breeding of livestock have crafted distinct breeds, altering genetic variation patterns within their genomes [150]. Strong selection has solidified advantageous mutations reflective of breed characteristics, productivity, or domestication traits, which results in selective sweeps where diversity is diminished in regions surrounding the selected allele. Recent investigations have confirmed the presence of selective sweeps across various livestock genomes, including cattle, sheep, goats, and horses [150,151].

Collectively, these discoveries have paved the way for GS, a method that scales up MAS and utilizes vast numbers of genetic markers to estimate their effects collectively. By concurrently assessing the roles of multiple markers, GS fosters the development of innovative animal breeding programs and models for genotype evaluation. This methodology has become essential for advancing animal breeding initiatives and enhancing genetic differentiation and polymorphism analyses at the DNA sequencing level [122,140].

## 7. Candidate Genes Driving Economic Traits

### 7.1. Growth and Muscle Development

The examination of candidate genes associated with growth and muscle development has uncovered several key players that are essential for important traits in livestock. Among these, *GH* and *IGF1* are significant regulators of postnatal growth, metabolic efficiency, and muscle development, impacting growth rates and carcass yield (Appendix A).

Additionally, genes like *POU1F1* and *ESR1* highlight the influence of hormonal regulation. *POU1F1* is involved in modulating pituitary hormone secretion, which supports muscle and milk production, while *ESR1* mediates the anabolic effects of oestrogen on tissue growth. *CAV3* and *CCNB1* are also noteworthy for their contributions to muscle fibre formation and cell cycle progression, both of which affect muscle growth and carcass yield (Appendix A).

The identification of *LCORL* and other stature-associated loci (such as *PLAG1* and *SMAD2*) further illustrates the genetic control over body size, a key factor in meat production. Additionally, growth factors *PDGF* and *BMP* play roles in muscle regeneration and skeletal development, which are crucial for recovery and structural integrity. Genes such as *MSTN* for hypermuscling and *GH/IGF1* for metabolic efficiency are now prioritized in breeding programs to enhance carcass yield while balancing feed efficiency (Appendix A).

### 7.2. Reproduction and Fertility Traits

The exploration of candidate genes linked to reproductive biology has unveiled critical genetic regulators of fertility, gametogenesis, and reproductive efficiency in livestock. Pivotal to sex determination and gonadal development, *SRY* has emerged as a key driver of testis formation, while *FOXL2* plays an essential role in female reproductive tissue development. Identified as vital for oocyte maturation and prolificacy, *GDF9* and *BMPR1B* directly influence litter size and female fertility [152]. Insights into breeding cycles in mammals are provided by *MTNR1A*, which underscores the genetic basis of seasonal reproduction (Appendix A).

Linkages between gamete production and quality are established through *TPPP3* and *KDM4C*, which govern male gamete stability and spermatogenesis, respectively. Hormonal and signalling pathways are further elucidated by *PLCB1*, a regulator of GnRH signalling, and *AR*, which is critical for sperm vitality. Additionally, developmental regulators, like *HMGA2* and *IGF2BP2*, are associated with foetal growth and embryonic viability, while *NR6A1* proves indispensable for oocyte and embryonic development. Contributions to egg production and gamete maintenance are highlighted by *MAGI1* and *BIRC6* (Appendix A). Additionally, Figure 2 represents an example of candidate genes detected in small ruminants and their associated functions [14,110].

Briefly, such findings emphasize actionable genetic targets, such as *BMPR1B* for enhancing prolificacy or *MTNR1A* for optimizing seasonal breeding, that can refine reproductive management, improve fertility rates, and support advancements in livestock breeding programs through biotechnology (Appendix A).

### 7.3. Milk Production and Composition

An analysis of candidate genes related to milk production and composition reveals the complex genetic factors influencing lactation efficiency and nutrient profiles in livestock. Key findings highlight important genes, such as the Casein gene cluster (*CSN1S1*, *CSN1S2*, *CSN2*, *CSN3*), located on chromosome 6, which play a crucial role in milk protein synthesis and cheese yield. Variations in *CSN1S1* and *CSN3* significantly affect casein content and protein efficiency, directly impacting milk quality. Additionally, *POU1F1* is vital for regulating growth hormone expression, influencing mammary gland development and optimizing milk yield (Appendix A).

Other important genetic markers include *ABCG2*, which modulates lipid transport and affects milk fat composition, and *LEP* and *SLC27A1*, which regulate energy balance and fatty acid uptake, respectively. These genes collectively shape milk fat content and overall dairy productivity. Furthermore, *DGAT1* on chromosome 14 governs milk fat composition, with its variants impacting lipid formation in mammary cells (Appendix A).

The findings from broad genomic regions associated with milk traits underscore the polygenic nature of lactation efficiency. Integrating genetic insights into breeding strategies, alongside nutritional management, can amplify dairy productivity.

### 7.4. Fibre Production, Coat Colour, and Skin Sensitivity

The genetic architecture influencing fibre quality, coat colour diversity, and skin sensitivity in livestock is shaped by key candidate genes and SSs. Significant discoveries reveal critical molecular pathways essential to traits in fibre-producing breeds, like Angora goats and Ankara sheep. Genes such as *CUX1* and *PLOD3*, both located on chromosome 25, are essential for collagen synthesis and hair texture regulation; mutations here can impact fibre strength and elasticity. Additionally, *KRT81* on chromosome 17 is vital for hair fibre integrity, while *FGF5* and *FGF7* regulate hair follicle development and length, presenting targets for enhancing fibre yield (Appendix A).

Coat colour variation is primarily regulated by *MC1R* on chromosome 18, which governs pigmentation through eumelanin and pheomelanin pathways. Polymorphisms in *ADAMTS20* and *TYRP1* further influence melanocyte functionality. Skin sensitivity and health are similarly affected by genes like *DUSP22* and *TIMP3*, which help maintain skin homeostasis and protect against degradation. Dysregulation of genes such as *IRF4* may exacerbate sensitivity to environmental factors, highlighting the importance of these genetic elements in livestock health (Appendix A).

These findings indicate practical breeding implications, targeting genes like *CUX1*, *PLOD3*, and *KRT81* through MAS to improve wool quality and resilience. Moreover, using *MC1R* and *ADAMTS20* variants could enable breeders to select for desirable coat colours tailored to niche markets. Variants in *TIMP3* and *DUSP22* may serve as biomarkers for skin health, guiding genomic screening for breeds susceptible to dermatitis (Appendix A).

### 7.5. Disease Resistance, Heat Tolerance, and Stress Response in Livestock

Appendix A highlights key candidate genes linked to various adaptation traits in livestock, focusing on disease resistance, heat tolerance, and stress response. Notably, the gene *SOD1* is associated with thermoregulation in Bos indicus, underscoring its role in oxidative stress defence, which is vital for maintaining cellular integrity under heat stress conditions. *DNAJC8* appears multiple times as crucial for heat tolerance in different cattle breeds, emphasizing its importance in assisting protein folding and protecting cellular functions during high temperatures.

Additionally, the genes *NDUFB3* and *DIS3L2*, which are involved in mitochondrial respiration and RNA processing, respectively, indicate that metabolic efficiency and gene expression regulation are critical for adaptation to environmental changes, particularly in Holstein Friesians. *TLR4* shows consistent immune response associations across GWASs (*n* > 10,000 cattle; *p* < 10^−8^) [143], whereas *HSPH1* evidence remains preliminary (single-breed study, *n* = 120) [153]. Early QTL scans for heat tolerance (e.g., chr5:36.25–36.75 Mb) require validation via functional studies given the low marker density (5–10 SNPs/Mb) versus modern 600K SNP arrays [4,154].

Moreover, other important candidates, such as *CFTR*, maintain ion transport and homeostasis, which is essential for thermoregulation, while *AQP1* facilitates water transport, aiding in hydration under stress. The involvement of genes like *IL6* and *RGS3* reflects the intricate network of inflammatory and signalling pathways that enable livestock to cope with extreme environmental conditions.

The identification of these candidate genes provides new avenues for breeding programs aimed at enhancing heat tolerance and overall resilience in livestock populations, which is increasingly important in the context of global climate change. Understanding the roles of these genes can lead to improved animal welfare and productivity, ensuring sustainable livestock farming practices. While these genetic targets offer transformative potential, their practical implementation faces challenges, including gene interactions and ethical concerns.

## 8. Challenges in GS

GS’s roots can be traced back to discussions on single locus selection [155]. However, by the late 1990s, the focus shifted from identifying key loci to a statistical approach considering the entire genome [156]. This approach was first implemented in dairy cattle [157] and later expanded to other species, including pigs [158] and poultry [159], along with numerous plant breeding programs [160].

In recent decades, a significant milestone in livestock genomics has been the emergence of GS, stemming from mixed results in MAS. While identifying and selecting against damaging alleles of large effect demonstrated clear success [158,161,162], this approach proved less effective for complex traits [163,164]. The development of SNP genotyping chips, which offer extensive genome coverage, combined with advanced estimation methods, helped overcome the challenges of managing numerous markers simultaneously [16].

The SNP chip, a family of high-throughput array-based SNP genotyping methods [165], plays a critical role in GS and is a contender for the most impactful genomic technology. Its advantages include sufficient markers for genome-wide genotyping, cost-effectiveness, accuracy, and the generation of well-structured tabular data, unlike sequence data, which is more computationally intensive and complex to interpret. Routine analyses frequently utilize SNP chip data; for example, through linear algebra, SNP chip genotypes can be transformed into a genomic relationship matrix. This matrix can then serve as a variance–covariance matrix in linear mixed models [166], which are central to GS. An extensive technical literature outlines efficient model-fitting and data-incorporation strategies [167].

### 8.1. Applications of GS

The concept of employing wide genetic markers (WGMs) in animal breeding was initially proposed by Meuwissen et al. [168]. Traditional MAS techniques were based on a relatively small number of genetic markers identified through controlled experiments [169]. The number of molecular markers (MMs) used in genomic evaluations varies depending on the methodology applied. GS assumes that the genetic differences for all traits can be captured through these markers (Figure 3A). However, the polygenic effects included in the model may not be entirely explainable by genetic markers. GS should ultimately focus on utilizing polymorphism-defined genotypes to select for specific phenotypes [170].

GS has the potential to significantly enhance traits associated with genetic improvement achieved through conventional methods. Some critical traits, such as carcass characteristics and disease resistance, are often expensive or challenging to measure. Others can only be evaluated in one sex or at the end of an animal’s life cycle, such as milk production and composition. Modern approaches, including GS, can address these constraints in genetic improvement [171]. Furthermore, GS can promote the resilience of animal species by facilitating increased production, gradual adaptation, and improved resistance to diseases [172]. This includes breeding efforts aimed at resistance to diseases, parasites, and challenges like fly-strike and facial eczema [173]. Additionally, GS presents ethical benefits by potentially reducing the prevalence of disease susceptibility in future generations. Current research is exploring GS for crucial biological traits like methane emissions and feed efficiency, which are often costly to evaluate and hard to implement widely on farms; thus, GS serves as a promising alternative [173].

GS is reliant on genotypic, phenotypic, and pedigree data, which can offer new opportunities for genetic improvement in farm animals, enhancing meat production (Appendix A), milk production (Appendix A), and fibre production (Appendix A), along with traits that are challenging to manage using traditional methods, such as reproduction (Appendix A), breeding seasonality, longevity, meat quality, and carcass composition (Appendix A). The feasibility of applying GS to small ruminants has been assessed in French dairy goats [174], Australian mutton breeds [175], and French dairy sheep [176]. The methodologies of GS have been effectively applied in dairy cattle breeding programs, resulting in reduced generation intervals. Although generation intervals are shorter in sheep and goats compared with cattle and buffalo, further reduction is still beneficial. This could lead to increased genetic gains per year, reducing costs and enhancing productivity [171].

GS involves testing SNPs and their high-density effects by utilizing models that simultaneously fit each SNP and treat these effects as random variables (Figure 3B). Various Bayesian models have been enhanced to execute statistical estimations using the Monte Carlo Markov Chain (MCMC) methodology [177].

### 8.2. Methodology of GS

To execute GS effectively within any animal population, certain prerequisites must be fulfilled: (1) a large number of animals per genotype, (2) the availability of phenotype information specific to each genotype, and (3) the application of appropriate statistical methods for accurate and effective genetic prediction. This assumes that the breeding program is optimal in its approach to (a) establish an accurate genetic evaluation system for relevant phenotypes, (b) align breeding objectives with targeted traits, and (c) ensure the breeding strategy supports long-term sustainable genetic gain [178,179,180] (Figure 3C).

Generally, the collected data is utilized as a reference to formulate new statistical models for estimating the effects of SNPs on target traits. The outcomes will yield predictive equations for estimating genomic-estimated breeding values (GEBVs) [181]. In scenarios where accurate phenotypes are not available, GEBVs for new individuals can be calculated from prediction equations based on the SNP genotype data. The accuracy of GEBVs is influenced by the heritability of the traits in question and the population size [182].

### 8.3. Conceptual Evolution from MAS to GS

MAS targets major-effect QTLs using sparse molecular markers (e.g., <50 loci), assuming traits are controlled by a few genes with large impacts [143]. In contrast, GS operates on the polygenic paradigm, where traits arise from thousands of small-effect variants requiring high-density genome-wide markers (e.g., 50 K–800 K SNPs) to predict an individual’s total genetic merit (GEBV) [183]. This shift from localization (identifying causal loci in MAS) to prediction (estimating genome-wide breeding values in GS) enables accurate selection for complex traits without prior QTL mapping [184,185].

Unlike MAS, which assumes discrete QTL effects, GS treats all marker effects as random variables in a mixed model (e.g., y = Xβ + Za + e), where a represents the vector of SNP effects estimated simultaneously. This allows GEBVs to capture the aggregate contribution of all markers, even those with negligible individual effects, thereby outperforming MAS for polygenic traits, like feed efficiency or disease resistance [5,16,186].

### 8.4. Challenges Faced by GS

GS is currently successfully implemented across a wide range of species, yet numerous questions persist regarding its long-term effects and the accuracy of genomic parameter estimates. Key concerns involve the robustness of GWASs when applied to both small and large datasets, as well as the stability of genomic predictions over time. Much of the existing research has focused on the linkage disequilibrium between pairs of loci, often overlooking higher-level equilibria that can lead to complications such as phantom dominance and epistasis. The Bulmer effect is another consideration, resulting in a decline in additive variance, although enhancing recombination rates can rekindle new genetic variance [178,187,188].

Genomic information may introduce biases in genetic parameter estimates due to genomic preselection, which can significantly increase estimation costs attributed to the dense nature of genomic data [189,190]. To manage these computational burdens, it may be beneficial to retain genotypes only for the most critical animals and utilize estimation methods that employ algorithms capable of identifying dense blocks within sparse matrices. Interestingly, GWASs conducted with smaller genomic datasets often reveal numerous marker-trait associations, while analyses based on much larger datasets typically identify only a handful. Many existing GWAS tools rely on relatively simplistic models, potentially leading to misleading results. These models may suffice for extensive datasets where pseudo-phenotypes, like deregressed proofs, help mitigate important trait influences. To reduce artifacts in GWAS results from smaller datasets, it is advisable to utilize data encompassing all individuals, whether genotyped or not, alongside realistic models and approaches that account for population structure [95].

Recent advancements allow for the computation of *p*-values based on genomic best linear unbiased prediction (GBLUP), which enables the use of more complex models while being restricted to genotyped animals, as well as the implementation of single-step GBLUP that incorporates phenotypic data from ungenotyped individuals [191]. Stability is a critical aspect during non-genomic evaluations, where genetic predictions maintain consistency without new data despite lower prediction accuracies [192]. In contrast, genomic evaluations for connected animals tend to fluctuate, as all animals with genotypes impact each other’s evaluations. This variability can lead to sudden drops in the rankings of top animals in subsequent assessments, undermining confidence in genomic predictions [193]. Although correlations between successive genomic evaluations are generally high, outliers may exhibit discrepancies equivalent to one standard deviation. One potential remedy for the instability of genomic evaluations is to base selection decisions on groups of animals rather than on individuals [95].

While GS presents significant advantages, it encounters various challenges that must be addressed. The methodologies used in GS can be complex, requiring advanced statistical knowledge and computational resources, complicating its implementation [95]. For sustained effectiveness, GS requires the continuous expansion of datasets through ongoing genotyping and phenotyping of new animals in the breeding population. The financial investment required for genotyping and analysis can be a significant barrier, particularly for smaller breeding operations that may lack necessary funding [187,194]. Additionally, for traits with low heritability, there might be limitations in the reliability of breeding value predictions, impacting the overall success of GS initiatives. Ensuring that GS methodologies are effectively applicable across various breeds and species poses another challenge, especially within small ruminants. Ethical considerations surrounding the use of genetic manipulation technologies raise important questions regarding animal welfare and the potential impacts on genetic diversity over generations [195,196]. Lastly, there may be apprehension among consumers and producers regarding the acceptance of animals selected through genomic methods, particularly within traditional farming practices. Navigating these challenges is essential to fully leverage the potential of GS in animal breeding programs.

## 9. Future Directions

### 9.1. Progress in Genomic Research for Ruminant Livestock

Ruminant livestock play a pivotal role in global agriculture, providing essential products such as meat, milk, and wool. Recent genomic advancements have allowed researchers to delve deeper into the genetic underpinnings of these animals, leading to a better understanding of traits that impact agricultural productivity. Current genomic research focuses on key areas such as construction of reference genome assemblies, population genomics, and identification of functional genes or variants linked to significant phenotypic traits. These traits include meat and carcass quality, reproduction, milk yield, feed efficiency, and disease resistance. The introduction of graphical pangenomics and telomere-to-telomere (T2T) genome assemblies has the potential to revolutionize our comprehension of the domestication processes and the molecular mechanisms governing these economically vital traits [162,197].

### 9.2. Advances in Gene Editing Technologies

The field of gene editing has experienced remarkable developments, particularly with the rise of CRISPR/Cas9 technology, which has emerged as a leading method due to its efficiency and precision [25,29,81,198]. Unlike older technologies, such as zinc finger nucleases (ZFNs) or transcriptional activation-like effector nucleases (TALENs), CRISPR/Cas9 simplifies the gene-editing process by requiring only the design of a single guide RNA (sgRNA) tailored to the target DNA sequence [199]. Its extensive applicability in various organisms has catalysed innovation within agriculture, particularly in enhancing crop traits and livestock characteristics. More recent techniques, like base editors and prime editing, further expand the scope of genetic modifications; these systems can facilitate precise edits with the potential to significantly enhance agricultural productivity [199,200,201], though their real-world impact remains contingent on regulatory approval.

### 9.3. Enhancing Genetic Diversity and Local Breed Resilience

Maintaining sufficient genetic variation within livestock populations is crucial for adapting to evolving climatic conditions and consumer preferences. However, the reduction in genetic diversity, particularly with the decline of smaller local breeds in favour of high-output international breeds, poses significant challenges. It is imperative to focus on improving the genetic potential of these local breeds, as they play an important role in sustainable agriculture [202]. By applying existing genetic improvement technologies in collaboration with farmers, we can revitalize local breeds. Breeding strategies that promote cooperation and centralized decision-making can be particularly effective when implemented for small breeds. Addressing the balance between productivity and genetic diversity will be essential for ensuring the long-term survival of local livestock breeds amid changing agricultural landscapes [203].

### 9.4. Super-Pangenomes for Precision Breeding

Constructing transgenus pangenomes incorporating diverse breeds and wild relatives will uncover SVs linked to adaptation and productivity. Initiatives like the Bovine Pangenome Consortium aim to resolve hidden genomic diversity, enabling precise mapping of genes influencing traits such as heat tolerance and muscle development [55,204,205].

### 9.5. Multi-Omics Integration for Trait Dissection

Combining genomic, transcriptomic, and proteomic data will elucidate regulatory networks underlying complex traits. For instance, *IGF2* [206] and *BMP15*’s [152] roles in muscle growth and fertility, respectively, can be validated through multi-omics pipelines, accelerating marker-assisted breeding [76]. Public data repositories, such as FarmGTEx, will drive meta-analyses to identify novel QTLs.

## 10. Ethical and Regulatory Imperatives

Getting ground-breaking genetic tools from labs into real-world use faces more than just technical challenges; it is tangled in deep public scepticism and a messy global regulatory web. People are not sold on gene-edited food, with nearly two-thirds in the EU wary due to its “unnatural” feel and fears of corporate control [207]. Ethical concerns also flare around animal welfare, like CRISPR-edited Belgian Blue cattle suffering painful births from extreme muscling [208] or pigs engineered for muscle growth raising similar welfare alarms [209]. Regulations vary wildly: the US treats edited animals like vet drugs (seen in GalSafe™ pigs’ 3-year, costly review [210]), while the EU groups gene editing with GMO, demanding strict rules and heavy tracking and labels [211,212]. Contrastingly, nations like Nigeria and Kenya fast-track edits using an organism’s own DNA, speeding up vital work, like heat-tolerant crops [213,214,215]. Moving forward responsibly demands a “One Welfare” approach balancing innovation with biodiversity, fairness, and proper compensation for those sharing genetic resources [213,214,215].

## 11. Conclusions

To meet the escalating global demand for animal products, all while navigating growing environmental and ethical challenges, we urgently need innovative approaches to livestock breeding. This review synthesizes evidence from 1455 studies spanning seven decades, highlighting the critical roles of candidate genes (e.g., *IGF2*, *DGAT1*, and *MSTN*), QTLs and GS in enhancing key traits, including muscle development, milk composition, disease resistance, and climate resilience. Advanced molecular tools, such as high-throughput sequencing, GWASs, and CRISPR-Cas9 editing, demonstrate how precision breeding accelerates genetic gains and improves sustainability. For instance, GS substantially reduces generation intervals and enhances traits like feed efficiency and thermotolerance, while CRISPR-mediated modification of the *PRNP* gene conferred scrapie resistance in goats, and SNP-based selection increased milk yield in heat-stressed tropical cattle. Despite progress, significant research gaps persist: the interplay between genotype and environment requires deeper elucidation, and complex epistatic interactions warrant further investigation. Ethical concerns regarding genetic diversity loss demand immediate attention, particularly given the 17% decline in livestock breeds since 2000. Conservation strategies for vital genetic reservoirs, such as climate-adapted Sahiwal cattle and parasite-resistant Red Maasai sheep, and their integration into breeding programs need refinement. Future efforts necessitate integrated multidisciplinary approaches, combining genomics with phenomics, metabolomics, and advanced biostatistics to decode complex traits. Initiatives like the Bovine Pangenome Consortium and multi-omics frameworks are essential to advance understanding of challenging attributes like methane mitigation and parasite resistance, whose underlying mechanisms and optimal breeding strategies remain poorly characterized. Robust ethical frameworks must also be established to balance production efficiency, biodiversity conservation, and animal welfare, with clear guidelines for reconciling potential conflicts. Collectively, this work provides a roadmap for researchers, breeders, and policymakers to develop livestock systems that are economically viable, climate-resilient, and capable of responsibly meeting 21st century food security imperatives through synergistic innovation and conservation.

## Figures and Tables

**Figure 1 ijms-26-07688-f001:**
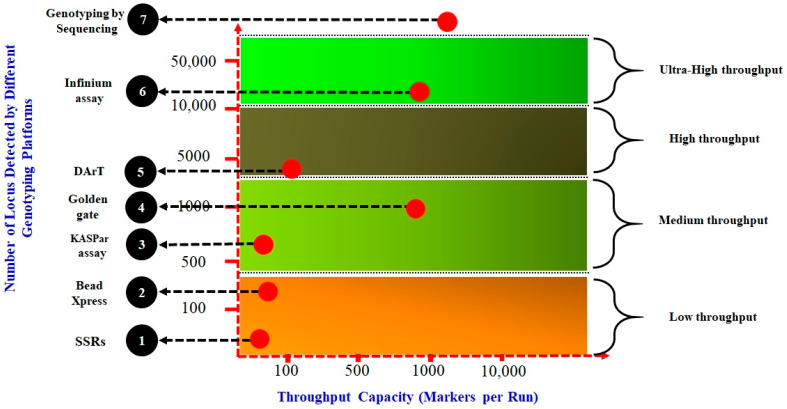
Genotyping platforms categorized by throughput capacity: This illustrates various genotyping marker assay platforms categorized from low to ultra-high throughputs. The platforms include the following: (1) Simple Sequence Repeats (SSRs), which are tandem repeats of short DNA motifs typically found throughout the genome and are influential in genetic diversity studies; (2) BeadXpress, which leverages VeraCode technology for flexible and efficient SNP scanning; (3) Kompetitive Allele Specific PCR (KASP), a fluorescence-based method that allows for allele-specific genotyping; (4) Golden Gate assay, known for its high multiplexing capabilities; (5) Diversity Arrays Technology (DArT), which enables simultaneous assessment of numerous markers via DNA hybridization; (6) Infinium assay, an ultra-high-throughput SNP genotyping method that can analyse millions of SNPs in a single sample; and (7) Genotyping by Sequencing (GBS), a NGS technique that allows for high-throughput SNP discovery and genotyping by digesting and sequencing.

**Figure 2 ijms-26-07688-f002:**
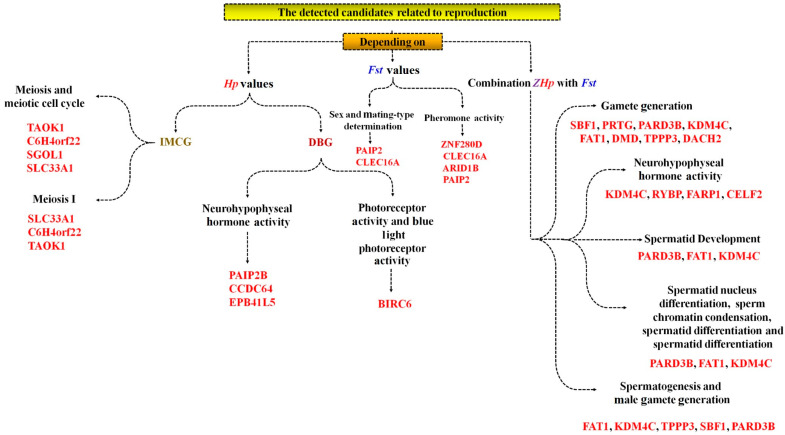
Candidate genes influencing reproduction in small ruminants of the Inner Mongolia cashmere goat breed (IMCG) and the Dazu black goat breed (DBG) based on insights from whole-genome sequencing studies.

**Figure 3 ijms-26-07688-f003:**
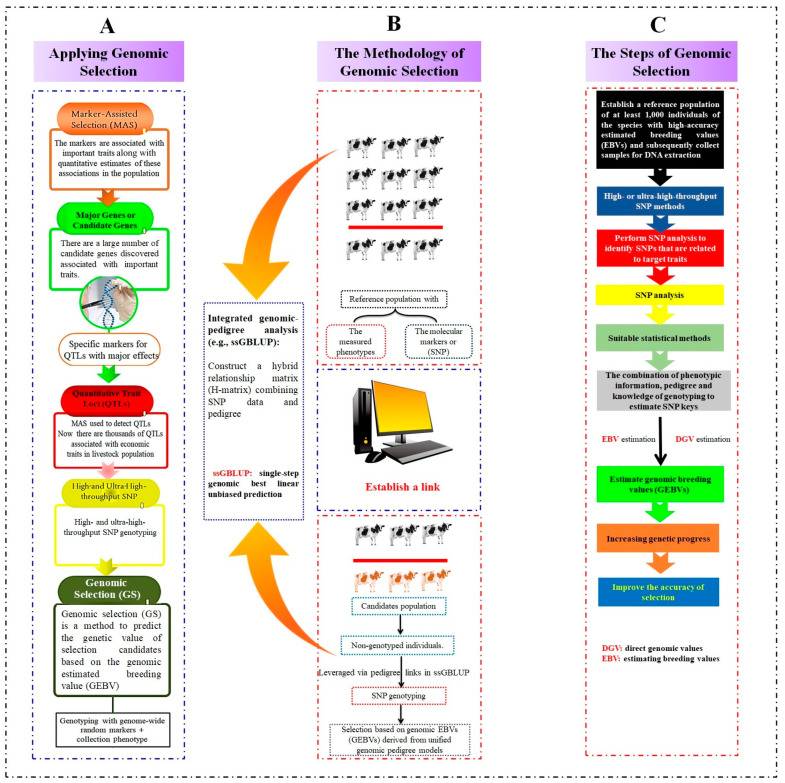
Genomic selection (GS): A paradigm-shifting method that predicts the total genetic merit using genome-wide markers. (**A**) Application of genomic selection: Illustrates the practical applications of genomic selection. (**B**) Methodology of genomic selection: Describes the core methodologies employed in genomic selection. (**C**) Steps in genomic selection: Outlines the sequential steps in the genomic selection process.

**Table 1 ijms-26-07688-t001:** Comprehensive overview of research studies on livestock genetics and breeding technologies used in this review.

Platform	Total Studies	Year Range	Dominant Topics
Elsevier	285	1988–2025	Genetic diversity, mitochondrial DNA analysis, gene polymorphisms (e.g., *BMP15*, *GDF9* and *PRLR*), ovarian follicular development, milk protein synthesis (caseins, β-lactoglobulin), heat shock proteins (HSPs), *IGF* signalling, coat colour genetics (*MC1R* and *KIT*), QTL mapping, and CRISPR applications.
Springer	142	1987–2025	GS, domestication history, mitochondrial genomics, *MHC-II* diversity, heat stress adaptation, prion protein (*PRNP*) polymorphisms, GWAS for disease resistance, microbiome interactions, and transcriptome analysis.
MDPI	89	2011–2025	Molecular tools (SNPs, CRISPR-Cas9), epigenetics (DNA methylation), cashmere fibre traits (KAP genes), immune response genes (*PTX3* and *TLRs*), gut microbiota studies, and GWASs for litter size and growth traits.
Frontiers	47	2016–2025	Coat colour genetics (FGF5 and TYRP1), immune system regulation, genomic prediction accuracy, CRISPR-mediated gene editing, climate resilience in livestock, and microbiome–host interactions.
Wiley	35	1994–2025	Follicle-stimulating hormone (*FSH*), GnRH receptors, candidate gene studies (*POU1F1* and *MSTN*), lactation traits, keratin-associated proteins (*KAPs*), and myostatin (*MSTN*) polymorphisms.
BioMed Central	28	2001–2025	Mitochondrial genome diversity, SNP discovery, gene expression profiling (RNA-Seq), microbiome dynamics, and functional annotation of genomic regions.
PLOS	19	2009–2025	Genome-wide selection signatures, CRISPR applications in disease resistance, comparative genomics, parasite resistance (Haemonchus contortus), and transcriptomics under heat stress.
Oxford University Press	15	1998–2025	MHC class I/II evolution, phylogenetic studies, immune gene polymorphisms, and livestock adaptation to tropical environments.
Taylor & Francis	12	2005–2025	Candidate gene association studies, prolactin (*PRL*) gene variants, reproductive traits (litter size), and milk yield optimization.
Nature Research	9	2002–2025	Whole-genome sequencing (WGS), domestication genomics, functional studies of growth hormones (*GH* and *IGF1*), and evolutionary biology of ruminants.
Additional Sources	755	1951–2025	Foundational studies (domestication history, SNP surveys), breed-specific trait analysis (e.g., Booroola fecundity gene), disease resistance (*SPP1*, osteopontin), mitochondrial haplogroups, conference proceedings, and institutional reports, FAO/UN publications.

**Table 3 ijms-26-07688-t003:** Comprehensive summary of documented quantitative trait loci (QTLs) in livestock and aquaculture species.

Species	Total QTLs (eQTLs/SNPs)	Publications	Genome Builds	Base Traits	Trait Variants	Key Traits Influenced
Cattle	193,453	1206	5	558	417	Growth, milk yield, disease resistance, and reproduction
Pig	57,041	854	3	406	1088	Meat quality, litter size, fat deposition, and immunity
Chicken	29,116	416	4	246	246	Egg production, growth rate, and feed efficiency
Sheep	5417	289	4	178	264	Wool quality, parasite resistance, and body size
Horse	2482	129	2	71	14	Athletic performance, coat colour, and skeletal traits
Goat	2713	47	2	90	120	Fibre quality, milk traits, and disease resistance
Rainbow Trout	2201	23	2	35	6	Growth rate, disease resistance, and stress tolerance

**Table 4 ijms-26-07688-t004:** Supported genome assemblies for QTL data alignment (as of April 2025).

Species	Assembly Name	Breed/Strain	Accession Numbers	Key Features	Ref.
Cattle	ARS_UCD1.2	Hereford	GCA_002263795.2	7× coverage, the combination of sequencing technologies	[123]
	ARS_UCD2.0	Hereford	GCA_002263795.4	31 chromosomes, 37,073 genes	
	Btau4.6	Hereford	GCA_000000095.4	7-fold mixed assembly	[63,124]
	Btau5.0	Hereford	GCA_000003205.6	95% genome coverage	
	UMD3.1	Hereford	GCA_000001245.5	Celera Assembler	[125]
Chicken	GG4.0	Red Junglefowl	GCA_000002315.2	Initial draft assembly	[126]
	GG5.0	Red Junglefowl	GCA_000002315.3	70× PacBio coverage	[127]
	GRCg6a	Red Junglefowl	GCA_000002315.5	80× SMRT sequencing	[127]
	GRCg7b	White Leghorn	GCA_016699485.1	Latest assembly	[128]
Goat	CHIR1.0	Yunnan Black	GCA_000317765.1	Initial assembly	-
	CHIR_ARS1	San Clemente	GCA_001704415.1	50× PacBio coverage	[41]
Horse	EC2.0	-	GCA_000000165.1	Initial draft	[64]
	EC3.0	-	GCA_002863925.1	88× coverage	
Pig	SS10.2	Duroc	GCA_000003025.4	Initial assembly	[62,129]
	SS11.1	Duroc	GCA_000003025.6	65× PacBio reads	
	SS_MARC1	Cross-bred	GCA_002844635.1	Landrace–Duroc–Yorkshire	-
Rainbow Trout	OM1.0	Swanson	GCA_002163495.1	Male genome	[130]
	OM1.1	-	GCA_013265735.3	Doubled haploid	
Sheep	OAR3.1	Texel	GCA_000298745.1	75× Illumina	[44]
	OAR4.0	Texel	GCA_000298745.2	Improved annotation	[61]
	OAR_rambo1	Rambouillet	GCA_002742125.1	126× coverage	
	OAR_rambo2	Rambouillet	GCA_016772045.1	Latest assembly (includes PacBio reads)	-

## Data Availability

All data generated or analysed during this study are included in this manuscript, its information files, and additional files: Appendix A.

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
