# Peer review of "Candidate Genes, Markers, Signatures of Selection, and Quantitative Trait Loci (QTLs) and Their Association with Economic Traits in Livestock: Genomic Insights and Selection"

_ijms, 2025, doi:10.3390/ijms26167688_

Round 1

Reviewer 1 Report

Comments and Suggestions for Authors

Dear authors,

Thank you for submitting your comprehensive review article, Candidate Genes, Markers, Signatures of Selection and Quantitative Trait Loci (QTL) and Their Association with Economic Traits in Livestock: Genomic Insights and Selection. This article provides a timely and valuable synthesis of advances in livestock genomics and addresses critical challenges in breeding for productivity, disease resistance and climate resilience.

Below, I offer constructive feedback to further improve your manuscript.

Consider including a glossary of key terms. (e.g., "gene interactions" instead of "epistatic interactions") and define acronyms when first used.

Some concepts (e.g. CRISPR applications, GWAS restrictions) are revisited multiple times, which interrupts the narrative flow. Consolidate redundant discussions and use subheadings to improve structure (e.g., "CRISPR: Current applications and future directions"). Expand this section to include examples (e.g. how different countries regulate CRISPR-modified animals).

Ethical aspects are mentioned, but only briefly discussed. A more in-depth examination (e.g. public perception, regulatory hurdles for genetically modified livestock) would add value.

The final section would benefit from a clearer summary of research gaps and future directions, what is still unknown, and where should the field move next?

Author Response

Dear Reviewer (1), We sincerely thank you for the valuable feedback and constructive insights. Your comments have been instrumental in enhancing the clarity, rigor, and overall quality of our manuscript. We appreciate your time and expertise in helping us improve our work. Please see the attachment (Point-by-Point response to the two reviewers and editors). 

Notes:

           We have provided two copies of the manuscript.

  1. MS Clean version (with the main submission)
  2. MS_with Tracking version (as PDF files).

We've highlighted the changes in the manuscript as follows: those suggested by the handling editor are in blue, Reviewer #1's suggestions are in yellow, and Reviewer #2's are in green.

Additionally, we have uploaded a file containing our responses to the editor and reviewers, addressing each point individually.

    Sincerely yours,

        Authors

Reviewer 2 Report

Comments and Suggestions for Authors

Overall Assessment and Recommendation

This review manuscript undertakes a task of immense scope and importance. The authors have made a commendable effort in compiling an extensive body of literature, citing 1,455 studies across seven decades, which showcases the vast landscape of livestock genetics. The breadth of information, particularly on modern genomic tools, gives this work the potential to be a valuable reference. However, the manuscript is not suitable for publication in its current form. Its primary deficiencies are a disorganized structure and a lack of critical synthesis. The work currently reads as a comprehensive but disjointed compilation of information, rather than an insightful and integrated academic review. The fragmented narrative leads to significant redundancy, which detracts from its scholarly value and readability.

Recommendation: Major Revision. Given the manuscript's potential and the clear effort invested by the authors, I encourage a fundamental restructuring based on the key points below. If these issues are successfully addressed, the manuscript could become a high-impact contribution to the field.

Major Comments

  1. Structural Reorganization: A Shift from "Method-Listing" to "Problem-Driven" Narrative is Essential
  • The Problem: The manuscript's most critical flaw is its structure. It appears to be organized along two parallel axes simultaneously: one by methodology (e.g., QTL mapping, Genomic Selection) and the other by biological trait (e.g., Productivity, Reproduction). This dual structure is the root cause of constant repetition. Key concepts and genes (e.g., DGAT1, MSTN) are introduced and re-explained in multiple sections, preventing the reader from forming a coherent understanding. For example, the concept of Genomic Selection (GS) is discussed in both the introduction and again at length in Chapter 8. This transforms the paper into a series of lists rather than a cohesive review.
  • Suggested Solution: The manuscript requires a complete reorganization to follow a “problem-centric” narrative, structured around the biological traits themselves. I suggest the following framework:
    • Introduction: Briefly outline the field's background, core challenges, and the role of genomics.
    • Main Chapters by Trait Category:
      • Genetic Dissection of Production Traits (Meat, Milk, Fiber): In this single chapter, integrate all relevant findings—from QTLs and candidate genes to selection signatures and the application of GS for these specific traits.
      • Genetic Basis of Adaptation and Health Traits: Follow the same integrated model for disease resistance, climate resilience, etc.
      • Genetic Regulation of Reproductive Traits: Consolidate all relevant research on fertility and litter size here.
    • Discussion & Outlook: Synthesize findings across trait categories, discuss methodological limitations, and propose future directions.

This reorganization will eliminate redundancy and create a clear, logical flow where the tools of genomics serve to answer biological questions.

  1. Content Deepening: A Move from "Information Compiling" to "Critical Synthesis" is Needed
  • The Problem: The review is largely descriptive, listing findings without sufficient critique or synthesis. It fails to differentiate the quality of evidence (e.g., early, low-resolution QTL studies are presented with the same weight as robust, modern GWAS findings). Furthermore, the manuscript astutely identifies the concurrent challenges of "increasing production efficiency" and the "loss of genetic diversity," but it fails to analyze the inherent tension between these points—namely, that intensive selection is a primary driver of diversity loss, and tools like GS can accelerate this process if not managed carefully.
  • Suggested Solution:
    • Evaluate Evidence: When discussing key genes or QTLs, the authors must comment on the strength of the supporting evidence, distinguishing between preliminary associations and well-validated findings.
    • Address Core Tensions: The manuscript should directly confront the central challenge of balancing production gains with the conservation of genetic diversity. It should analyze how modern genomic tools can be applied to both goals.
    • Identify Knowledge Gaps: Each major section should conclude by explicitly stating what remains unknown and what the key questions for future research are. This would elevate the work from a summary to a forward-looking guide for the field.
  1. Figure and Table Quality: Visuals Must Be Refined to Support the Core Arguments
  • The Problem: The visual aids do not effectively support the text and, in some cases, detract from its quality. Figure 2 (pathway map) is disconnected from the manuscript's content, including many genes not discussed in the text, with its origin and purpose unclear. Figure 4 (GS/MAS flowchart) is convoluted and difficult to follow. The candidate gene lists in Tables 6-10 are overly comprehensive, making it difficult to distinguish high-confidence genes from those with weaker evidence. I suggest highlighting the most strongly validated candidates to improve focus. Separately, Table 5 contains editorial and formatting errors. The most notable is a duplicated entry for the "CHIR1.0" genome assembly for the Yunnan Black goat, which appears twice. The redundant entry in the final row should be removed for accuracy.
  • Suggested Solution:
    • Figures: I strongly recommend deleting Figure 2 or completely redesigning it to be a schematic that illustrates a key pathway using only genes discussed in the text. Figure 4 should be simplified, perhaps by splitting it into separate, clearer diagrams.
    • Tables: In the main text, use smaller, summary tables that highlight only the most important and well-supported genes. Move the exhaustive gene lists to Supplementary Materials.
    • Crucially, add a "Strength of Evidence" column to the tables (e.g., Grade A for functionally validated genes; Grade B for strong GWAS hits; Grade C for preliminary associations). This single change would dramatically increase the scientific value and utility of the tables.

Minor Comments

  • Section Formatting: The "Materials and Methods" section is not standard for a narrative review. This should be revised to a brief "Scope and Literature Search Strategy" paragraph.
  • Precision of Terminology and Concepts:The manuscript's use of terminology is sometimes imprecise, and it fails to clearly distinguish between related concepts. For example, the relationship between Marker-Assisted Selection (MAS) and Genomic Selection (GS) is oversimplified, with Figure 4 describing GS as a "big scale version from (MAS)". This fails to capture the fundamental theoretical difference between them: MAS relies on a few molecular markers tightly linked to a QTL, typically targeting major or moderate-effect genes. In contrast, GS is based on the premise that complex traits are controlled by numerous small-effect QTLs, and therefore requires genome-wide markers to estimate an individual's total genetic value (GEBV). The key is "prediction," not "localization." At the beginning of Section 8, the authors should include a paragraph that clearly explains the conceptual evolution and theoretical differences from MAS to GS to help readers build an accurate framework.
  • Supplementary Materials:The current Supplementary File 1 is merely a list of the 1455 references, which has limited value as supplementary material. As mentioned earlier, the supplementary materials would be the ideal place for the exhaustive candidate gene tables (i.e., the full versions of the current Tables 6-10). This would keep the main text concise and readable while providing comprehensive data for interested readers.

Conclusion

The authors have assembled an impressive and valuable collection of information. However, to be publishable, this information must be reshaped into a coherent, critical, and well-structured narrative. A major revision focusing on the structural and analytical points raised above is necessary. I believe that if these revisions are undertaken, the manuscript will be transformed from a detailed compilation into an excellent review with significant guiding value for the livestock genetics community.

Author Response

Dear Reviewer (2), We are deeply grateful for your insightful feedback and constructive suggestions. Your thorough review has significantly contributed to improving the clarity and quality of our manuscript. We greatly value your expertise and the time you dedicated to helping us enhance our work. Thank you sincerely for your support.

Notes:

           We have provided two copies of the manuscript;

  1. MS Clean version (with the main submission)
  2. MS_with Tracking version (as PDF files).

We've highlighted the changes in the manuscript as follows: those suggested by the handling editor are in blue, Reviewer #1's suggestions are in yellow, and Reviewer #2's are in green.

Additionally, we have uploaded a file containing our responses to the editor and reviewers, addressing each point individually.

    Sincerely yours,

        Authors

Round 2

Reviewer 2 Report

Comments and Suggestions for Authors

The authors have undertaken a major revision of their manuscript in response to my previous critique. I commend the authors for their diligent and comprehensive effort in restructuring the manuscript and addressing the core deficiencies identified in the first round of review. The revised manuscript is a significant improvement, transitioning from a "disjointed compilation" to a much more "coherent, critical, and well-structured narrative". The authors have correctly identified and implemented a problem-centric, trait-driven structure, which has successfully eliminated most of the redundancy that plagued the original draft. The work now reads with a clear, logical flow, making it a much more valuable reference for the field. However, while the structural and high-level issues have been largely resolved, the manuscript still requires further refinement to reach the standard for publication. Several areas of the text, particularly in the details and the supporting visuals, show inconsistencies or a lack of thoroughness in the revision process.

Recommendation: Minor Revision. The manuscript is now close to being a high-impact contribution, but a final targeted revision is needed to polish the details and ensure all comments from the first review have been fully implemented.

(1)Figure 4 (now Figure 3): The authors state that Figure 4 was revised for clarity and accuracy. However, the new Figure 3 still contains problematic language. The text in the left panel (A) still refers to Genomic Selection (GS) as "a big scale version from (MAS)". The authors' response in R2(6) acknowledges that this phrase is an oversimplification and that GS and MAS are conceptually different. While a new paragraph in Section 8.3 was added to explain this distinction in the text, the visual aid itself remains uncorrected. This is a significant inconsistency that must be fixed. The figure caption was changed, but the image text remains misleading.

(2)Tables: The authors have moved the exhaustive gene lists (Tables 6-10) to the Supplementary Materials, which is a key improvement. This makes the main text more concise and readable. However, the response does not confirm whether a "Strength of Evidence" column was added to the new supplementary tables as suggested. Adding this column would significantly increase the scientific value and utility of the data for readers. Please confirm that this was done and, if not, please implement this change.

(3)Table 5 (now Table 4): The authors' response notes that Table 5 (now Table 4) contained a duplicated entry for the "CHIR1.0" genome assembly. The revised Table 4 still contains this duplicated entry in the final row. This must be corrected for accuracy.

Author Response

Dear Reviewer (2), We are deeply grateful for your insightful feedback and constructive suggestions. Your thorough review has significantly contributed to improving the clarity and quality of our manuscript. We greatly value your expertise and the time you dedicated to helping us enhance our work. Thank you sincerely for your support.

Notes:

  A. We have provided two copies of the manuscript.

  1. MS Clean version (with the main submission)
  2. MS_with Tracking version (as PDF files).

B. We've highlighted the changes in the manuscript as follows: those suggested by Reviewer #2 are in yellow.

C. Additionally, we have uploaded a file containing our responses to the editor and reviewers, addressing each point individually.

    Sincerely yours,

        Authors
